# Aluminium-induced component engineering of mesoporous composite materials for low-temperature NH₃-SCR

Ge Li[1,4], Baodong Wang [1,4✉], Ziran Ma[1], Hongyan Wang[1], Jing Ma[1], Chunlin Zhao[1], Jiali Zhou[1], Dehai Lin[1], Faquan He[1], Zhihua Han[1], Qi Sun [1,2✉] & Yun Wang [3✉]

Supported Mn₂O₃ is useful in achieving high dinitrogen selectivity at low temperature during ammonia-selective catalytic reduction (SCR). However, its controlled synthesis is challenging when the supporting material is the conventional pure silicon SBA-15 mesoporous molecular sieve. Here we show that silicon and aluminium in fly ash, the solid waste produced by coal-fired power plants, can be used to synthesize an Al-SBA-15 mesoporous molecular sieve support, which can guide the growth of Mn₂O₃ in the as-synthesized Fe-Mn/Al-SBA-15 NH₃-SCR catalyst. Its superior catalytic performance is demonstrated by the high NOₓ conversion (≥90%) and selectivity (≥86%) at low temperatures (150–300 °C). The combined theoretical and experimental results reveal that the introduction of Al induces the growth of Mn₂O₃ catalysts. Our findings, therefore, provide a strategy for the rational design of low-temperature NH₃-SCR catalysts through dopant-induced component engineering of composite materials.

[1] National Institute of Clean-and-Low-Carbon Energy, Beijing 102211, China. [2] Institute for Sustainable Energy and Resources, Qingdao University, Shandong 266071, China. [3] Centre for Clean Environment and Energy, Gold Coast Campus, Griffith University, Southport, QLD 4222, Australia. [4] These authors contributed equally: Ge Li, Baodong Wang. ✉email: baodong.wang.d@chnenergy.com.cn; qisun_l@hotmail.com.com; yun.wang@griffith.edu.au

The large amount of nitrogen oxides ($NO_x$) produced by the combustion of fossil fuels (mainly coal, oil, and natural gas) is an important precursor of fine particles and an important cause of smog[1]. China's smog pollution has had a serious impact on the global environment[2,3]. Ammonia-selective catalytic reduction ($NH_3$-SCR) of $NO_x$ is technically an efficient way to control and reduce stationary $NO_x$ emissions, such as from power plants, industrial boilers, and kilns. However, the conventional commercial $V_2O_5$-$WO_3$/$TiO_2$ catalyst is incapable of reducing $NO_x$ by $NH_3$ when the flue gas temperature is ≤300 °C[4–6]. Moreover, the key active ingredient V, a toxic and harmful element to the environment and human health, might be leached out to further pollute the environment. It is imperative to develop an environmentally friendly catalyst to efficiently reduce $NO_x$ at low temperatures.

In recent years, low-cost and nontoxic zeolite catalysts have become a favorable SCR catalyst carrier[7–12]. Among all the studied zeolites, SBA-15 mesoporous molecular sieves (MMS) have attracted the most attention due to their larger specific surface area (690–1040 m²/g), larger pore size (4.6–30 nm), and better hydrothermal stability[13–18]. In our previous work[19–21], Fe–Mn/SBA-15 catalysts were prepared using a fly ash-derived SBA-15 MMS as a support and tested for $NH_3$-SCR. The 11.2Fe-11Mn/SBA-15 catalyst exhibits high $NH_3$-SCR activities at 150–250 °C. However, this catalyst favors $N_2O$ formation. Previous studies reveal that the polymorph of $MnO_x$ including $MnO_2$, $Mn_5O_8$, $Mn_2O_3$, $Mn_3O_4$, and MnO is critical to the selectivity of the catalyst[22]. Among them, $Mn_2O_3$ has the best selectivity for the nontoxic $N_2$ production[22]. However, it is difficult to synthesize $Mn_2O_3$ on the SBA-15 MMS using impregnation method. Therefore, innovative pathways are needed.

From the molecular design point of view for the $NH_3$-SCR catalyst, the denitrification catalyst needs to have suitable acid-base and redox properties[23,24]. There are a large number of hydroxyl structures on the surface of SBA-15 MMS. Doping with Al can increase the acidity of the molecular sieves. The Brønsted (B) acid sites on the framework are generated, and a Lewis (L) acid is produced by extra-framework Al (EFAL) outside the pores. On the other hand, the surface can be loaded with active components to build the catalyst's redox properties. In our present work, we theoretically predicted that the introduction of Al could induce the formation of $Mn_2O_3$, as elucidated in detail later. However, efficiently doping Al into the framework of SBA-15 zeolites is difficult. In recent years, both one-step copolycondensation and post-synthesis methods have been employed to prepare Al-SBA-15 MMS[25–30]. In the one-step copolycondensation method, Al exists in the form of a complex, which is difficult to condense with the positively charged Si in the strong acid system. Therefore, it is challenging to successfully insert Al cations into the silica skeleton in this method. While more Al cations can be introduced in the post-synthesis method by strictly controlling the pH of the system and reducing the hydrolysis of the Al salt, a large amount of EFAL is swept away during the washing process, which reduces the total acidity of the Al-SBA-15 catalyst. Therefore, the development of an efficient way for effectively doping Al in SBA-15 with suitable acidity holds the key to the molecular design of high-performance $NH_3$-SCR catalysts.

In this study, fly ash, solid waste produced by coal-fired power plant, is used as the raw material for the preparation of SBA-15 MMS and $AlCl_3·6H_2O$ powders. An impregnation method is subsequently used to synthesize Al-SBA-15 MMS by controlling the pH value. After that, an impregnation method is used to prepare the Fe–Mn/Al-SBA-15 catalyst, which exhibits a high overall performance for $NH_3$-SCR denitrification at low temperature. In addition, the denitrification reaction mechanism for the Fe–Mn/Al-SBA-15 catalyst at 200 °C is investigated by in situ infrared (IR) spectroscopy. Our combined experimental characterization and computational results reveal that the superior performance of the Fe–Mn/Al-SBA-15 catalyst can be ascribed to the controllable growth of $Mn_2O_3$ polymorph due to the Al-induction.

## Results

**Characterizations of fly ash-derived Al-SBA-15 MMS**. Since the pH value is crucial for the aluminum salt hydrolysis and Si–O–Al bonds formation, we first investigated the effect of the pH value on the synthesis of fly ash-derived Al-SBA-15 MMS. 1 g of the fly ash-derived Si-SBA-15 MMS and 1 g of fly ash-derived $AlCl_3·6H_2O$ powder were added to 50 mL of absolute ethanol (pH = 4.7), 50 mL of $H_2O$ (pH = 3.5), 100 mL of $H_2O$ (pH = 2.6), and 150 mL of $H_2O$ (pH = 1.8), respectively. The impregnation experiment was carried out by magnetic stirring at 80 °C for 12 h. After spin-drying and calcination at 550 °C for 5 h at a heating rate of 5 °C/min, Al-SBA-15 MMS were synthesized. Figure 1a, b shows the small angle X-ray diffraction (SXRD) and X-ray diffraction (XRD) patterns of the Al-SBA-15 MMS prepared at different pH values. When the pH value was >3.5, the SXRD pattern of the obtained Al-SBA-15 MMS possesses a prominent peak at $2\theta = 0.8°$ and two small peaks at $2\theta = 1.6°$ and 1.8°, which have been indexed to the (100), (110), and (200) diffraction peaks of the SBA-15 XRD pattern, respectively[13]. This suggests that the original pore characteristic peak of the SBA-15 MMS maintains in this pH range. However, when the pH values were 2.6 and 1.8, the diffraction peak of the molecular sieve (100) crystal plane was obviously weakened, which indicated that the ordering of the molecular sieve pores was significantly reduced. The isoelectric point of $SiO_2$ is 1.7–3.5[28,29]. It suggests that the surface of $SiO_2$ is negatively charged when the pH value of the solution is higher than 3.5. Consequently, $Al^{3+}$ can be adsorbed through electrostatic interaction, which is beneficial for the formation of Si–O–Al bonds. The weakly acidic environment produced by the hydrolysis of inorganic aluminum salts (the pH value is close to the isoelectric point of silicon oxide) can also reduce the polymerization rate of silanol groups in the pore walls of SBA-15 material, which leads to a larger amount of silanol groups on the pore walls of the material. The silanol groups can interact with Al–OH in solution to form a Si–O–Al bond through high-temperature polymerization. Therefore, uniformly grafting Al to the silicon oxide mesoporous skeleton of the material is achieved when the pH value is higher than the isoelectric point of silicon oxide.

Figure 1c, d shows the small-angle powder XRD patterns and wide-angle XRD patterns of Si-SBA-15 before and after post-synthesis alumination under various Si/Al ratios. The XRD peaks were indexed to a hexagonal lattice with a $d_{100}$ spacing of 9.7 nm, which corresponded to a unit cell parameter $a_0$ of 11.2 nm by using the formula $a_0 = \frac{2d_{100}}{\sqrt{3}}$ [27]. The well-defined XRD patterns showed that all the samples retained the characteristic hexagonal mesostructure of SBA-15 after alumination. No $Al_2O_3$ diffraction peaks were observed in the wide-angle XRD patterns regardless of the amount of incorporated Al. This indicated that all the Al were incorporated into the SBA-15 skeleton or highly dispersed as non-skeleton Al. It, therefore, proves that Al-SBA-15 MMS have been successfully synthesized.

To avoid the influence of moisture on the surface hydroxyl test of the molecular sieves, we used in situ IR spectroscopy in the transmission mode to characterize the surface hydroxyl groups of SBA-15 and Al-SBA-15 MMS. Figure 1e shows that, compared with the SBA-15 MMS, the isolated silanol stretching vibration peak of the Al-SBA-15 (Si/Al = 4.17) MMS is obviously changed near 3740 $cm^{-1}$ with considerably small the peak intensity and

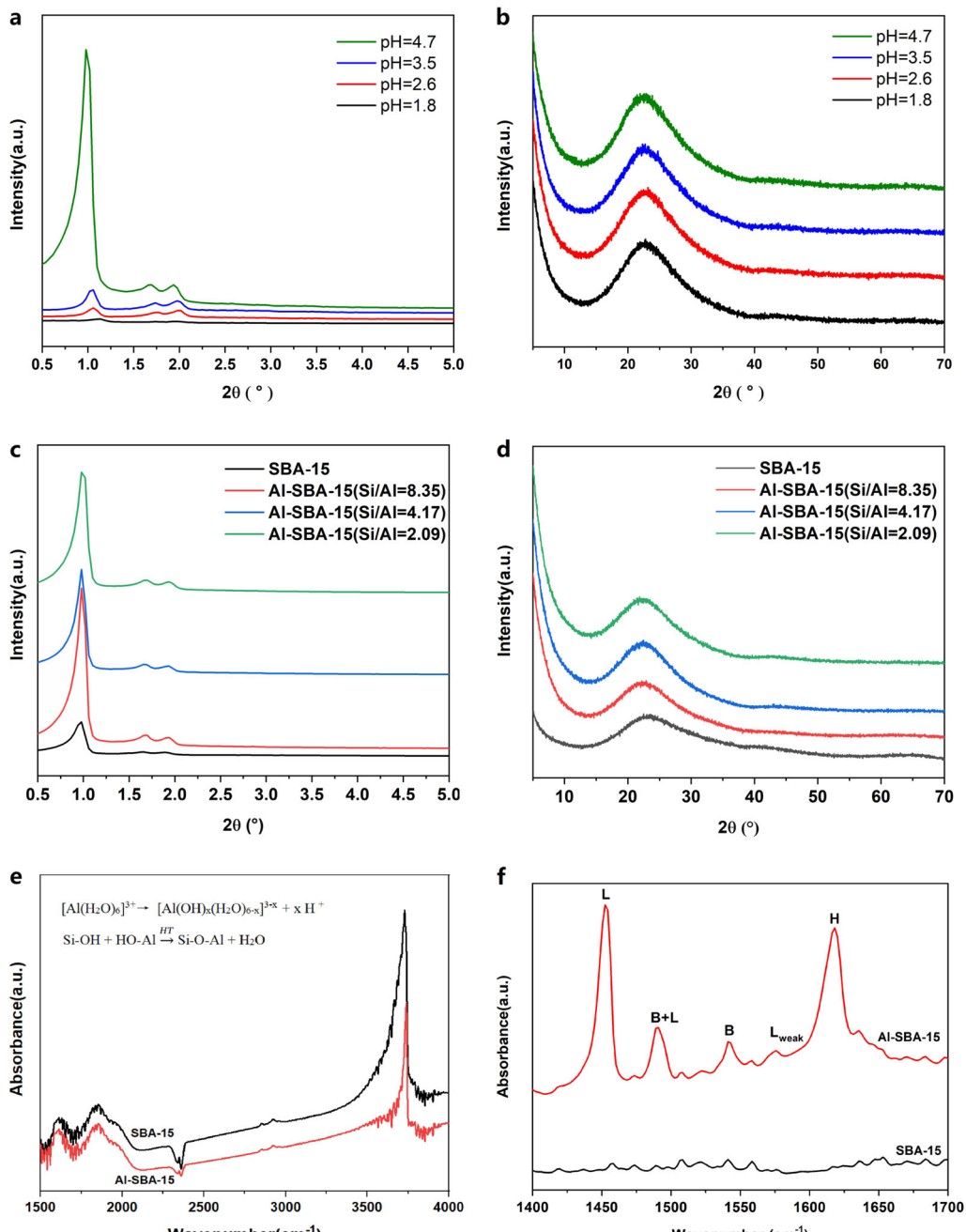

**Fig. 1 Structural characterization and IR spectra of Al-SBA-15 molecular sieves. a** SXRD patterns of Al-SBA-15 MMS prepared under various pH values. **b** XRD patterns of Al-SBA-15 MMS prepared under various pH values. **c** SXRD patterns of Al-SBA-15 MMS prepared under various Si/Al ratios. **d** XRD patterns of Al-SBA-15 MMS prepared under various Si/Al ratios. **e** In situ FTIR with transmission mode of SBA-15 and Al-SBA-15. **f** IR spectra of pyridine adsorbed on SBA-15 and Al-SBA-15.

peak area. This phenomenon suggested that Al doping modifies the surface silanol groups of SBA-15, as reported in the literature[27–29].

The FTIR spectra for pyridine (Py) adsorbed on SBA-15 and Al-SBA-15 (Si/Al = 4.17) are shown in Fig. 1f. The results reveal that pure Si SBA-15 is not acidic. After Al doping, Al-SBA-15 exhibits two strong absorption peaks at 1452 and 1618 cm$^{-1}$. The absorption peak at 1452 cm$^{-1}$ arose from the complex (LPy) produced by interactions of Py molecules with Lewis acids, and the peak at 1618 cm$^{-1}$ resulted from pyridine (HPy) adsorbed on surface hydroxyl groups through hydrogen bonds. The absorption peak at 1541 cm$^{-1}$ corresponds to Brønsted acidic centers.

The peak at 1490 cm$^{-1}$ corresponded to a combination of Lewis and Brønsted acids. The formation of Brønsted acidic centers was attributed to Si–OH–Al formed by polymerization of Al–OH produced by hydrolysis of the Al-containing solution (AlCl$_3$) and Si–OH in the pore walls of SBA-15 in the synthesis system. The Py-IR results demonstrate that the incorporation of Al can significantly enhance the surface concentrations of L and B acids to provide favorable conditions for its applications as a catalyst carrier.

$^{29}$Si and $^{27}$Al magic angle spinning (MAS) nuclear magnetic resonance (NMR) spectroscopies were used to investigate the Al-SBA-15 samples to confirm the effects of Al doping on silanol of

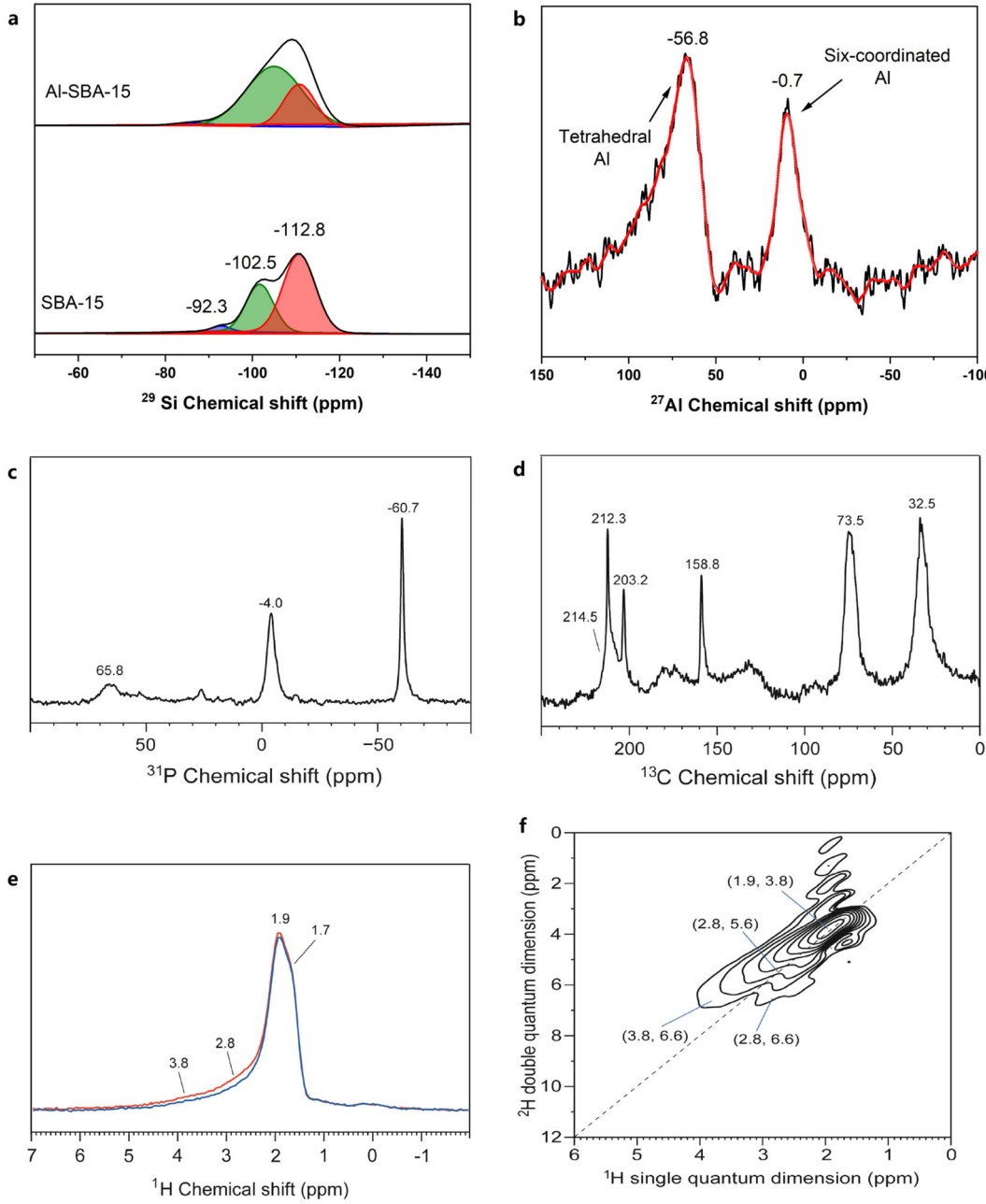

**Fig. 2 NMR spectra of Al-SBA-15 samples. a** $^{29}$Si MAS spectra (compared with SBA-15). **b** $^{27}$Al MAS spectra. **c** $^{31}$P MAS NMR spectra. **d** $^{13}$C CP/MAS NMR spectra. **e** $^{1}$H/$^{27}$Al TRAPDOR NMR spectra. **f** $^{1}$H DQ-MAS NMR spectra.

Si species in the samples. Figure 2a shows that the spectrum of Al-SBA-15 is significantly broadened, compared with the spectrum of SBA-15, indicating that Si-Al dipole action occurs[28]. The $^{29}$Si MAS NMR spectrum of SBA-15 showed three chemical shifts at $\delta = -92.3$, $-102.5$, and $-112.8$, which corresponds to Si(OSi)$_2$(OH)$_2$ (Q$^2$, chain structure), Si(OSi)$_3$(OH) (Q$^3$, rack structure), and Si(OSi)$_4$ (Q$^4$, island structure), respectively[29]. Q$^3$ is mainly on the surface of the tunnel and is important for material modification. The fractionated peak fitting results suggest that the proportion of Q$^3$ is significantly increased after Al doping, indicating that the surface Si–OH is significantly increased. When Al$^{3+}$ substitutes Si$^{4+}$ without changing the oxygen coordination number, a hydroxyl group on Al$^{3+}$ is required to maintain the charge neutral condition. Consequently, a Brønsted acidic center is generated. This is the reason for the increase in Si–OH after Al doping. The framework Si:Al ratio in

the MMS can be calculated from Q$^3$ and Q$^4$. The framework Si:Al ratio for the Al-SBA-15 MMS sample was calculated to be 14.64 by using the equation[28,29]:

$$\frac{\text{Si}}{\text{Al}} = \frac{\sum_{i=0}^{4} I_{\text{Si}[n\text{Al}]}}{\sum_{i=0}^{4} 0.25n I_{\text{Si}[n\text{Al}]}}. \tag{1}$$

The coordination environments of Al in the Al-SBA-15 materials were determined by $^{27}$Al MAS NMR spectroscopy. The spectra of the Al-SBA-15 materials show two resonances (Fig. 2b). The resonance at 0 ppm, which arises from six-coordinated Al, is more intense than the resonance at 56 ppm, which can be assigned to the tetrahedral framework Al formed in the mesoporous walls of the material[31]. This demonstrates that a post-synthesis procedure can be used to graft Al–OH species into

the mesoporous walls of SBA-15. And the formation of Si–O–Al bonds increases the relative amount of tetrahedral Al in the pore walls of the resulting material. These results combined with the data obtained by Py-IR spectroscopy suggest that Al atoms were successfully incorporated into the porous surface of the mesoporous SBA-15 structure by a post-alumination procedure.

The relative value of the $^{31}P$ chemical shift can be used to determine the acidic strength of a catalytic material. It is generally believed that the trimethylphosphine molecule (TMP) interacts with a Brønsted acid to form TMPH$^+$ ions, whose chemical shift is ~0 ppm in the single-pulse $^{31}P$ spectrum. Although the chemical shift of the TMP molecule adsorbed on a Lewis acid is between −32 and −58 ppm, the chemical shift of physically adsorbed TMP is approximately −60 ppm[32]. Figure 2c shows the single pulsed $^{31}P$ spectrum of TMP adsorbed on Al-SBA-15. Two signals were well resolved at −4.0 and −60.7 ppm. When the TMP was removed under vacuum at 80 °C for 2 h, the peaks at −4 and −60 ppm were still present, which indicates the presence of a stable Lewis acid in the Al-SBA-15 sample. This is consistent with the Py-IR results. The signal at 65.8 ppm showed that TMP was oxidized to TMPO adsorbed on the Brønsted acid of the Al-SBA-15 sample. Meanwhile, it was also indicated that the Al-SBA-15 sample prepared by the impregnation method had both Lewis acid and Brønsted acid. According to the integral area of the two peaks, the L acid had a stronger acidic strength than the Brønsted acid. The Al-SBA-15 sample prepared using the ion exchange method in previous studies[28,29] only had Brønsted acid caused by the skeleton Al. While the Al-SBA-15 sample prepared by the impregnation method contained non-framework Al.

Previous studies have confirmed that the isotropic chemical shift of the 2-$^{13}C$-acetone carbonyl carbon can be used to measure the strength of various solid acids[33,34]. After the molecular sieve adsorbs acetone, a stronger Brønsted acidity will result in a greater chemical shift to the lower field. Therefore, acetone is a useful probe molecule to measure the relative acidic strength of a solid acid. The $^{1}H/^{13}C$ CP MAS spectrum after adsorption of acetone by the Al-SBA-15 MMS is shown in Fig. 2d. The three peaks at 212.3, 73.5, and 32.5 ppm correspond to the products of dimerization or trimerization of acetone molecules, in addition to the peak at 214 ppm, which was considered as the signal of unreacted acetone adsorption at unequal acid sites[31]. Because the Al-SBA-15 MMS showed no peak at 232 ppm, this indicated that the Al-SBA-15 MMS had no more acidic acid sites. The Brønsted acidity of the signal at 218 ppm was also weaker than that of HZSM-5 (chemical shift was 223 ppm)[31]. According to the integrated area of the peak, the Lewis acid strength of the Al-SBA-15 MMS was greater than the acidic strength of the Brønsted acid, which is consistent with the results of TMP-$^{31}P$ NMR and Py-IR.

To establish a correlation between the various hydroxyl groups and Al, we performed $^{1}H/^{27}Al$ transfer of population in double resonance (TRAPDOR) experiments on the Al-SBA-15 samples. Figure 2e shows a weak signal at 2.8 ppm and a strong signal at 1.9 ppm, which were ascribed to AlOH groups and SiOH in the vicinity of Al atoms, respectively. This structure was similar to that of germinal or hydrogen-bonded hydroxyl groups[33]. The peak at 3.8 ppm was assigned to the protons at the Brønsted acid sites[33], which indicated that Al was tetrahedrally coordinated in the silica framework. Moreover, the concentration of the bridged hydroxyl groups was low with respect to Si–OH.

It is widely believed that the non-skeleton is mainly composed of aluminum oxide ions (such as [AlO]$^+$, [Al(OH)$_2$]$^+$, [AlOH]$^{2+}$) and some electrically neutral species (such as AlOOH, Al(OH)$_3$, and polymeric Al$_2$O$_3$)[33]. Whether there is a synergistic effect between the Brønsted acid associated with the framework aluminum and the Lewis acid outside the framework can be verified by two-dimensional $^{1}H$ double quantum (DQ)-MAS NMR experiments. Figure 2f shows a two-dimensional $^{1}H$ DQ-MAS NMR image of an Al-SBA-15 molecular sieve. Cross-peak pairs at (3.8, 6.6) ppm and (2.8, 6.6) ppm revealed the spatial proximity between the non-framework aluminum hydroxy (Lewis acid) and the framework Brønsted acid proton in the Al-SBA-15 MMS. At the same time, two sets of autocorrelation peaks (2.8, 5.6) ppm and (1.9, 3.8) ppm were observed, which indicated that the same hydroxyl groups were not isolated from each other but were spatially adjacent (distance less than 5 Å). The autocorrelation peak at (2.8, 5.6) ppm arose from extra-framework AlOOH or [Al(OH)]$^{2+}$ in close proximity or EFAL species containing more than one hydroxyl group, such as Al(OH)$_3$ and [Al(OH)$_2$]$^+$. The spatial proximity of the Lewis acid and the Brønsted acid increased the overall acidity of the catalyst, which is favorable for the SCR reaction.

Supplementary Fig. 1 shows that the N$_2$ adsorption–desorption isotherms of all the SBA-15 MMS, whether pure silica or Al-doped molecular sieves, were typical type IV isotherms with H1-type hysteresis loops. The pore volume, pore size, and Brunauer–Emmett–Teller (BET) specific surface area decreased with increasing amount of grafted Al. The details of the N$_2$ adsorption–desorption isotherms of all samples are shown in Supplementary Fig. 1, Supplementary Table 1, and Supplementary Note 1 of the Supplementary Information.

Figure 3 is the high-angle annular dark field scanning transmission electron microscopy (HAADF-STEM) images of Al-SBA-15 MMS, which evidences that the resulting Al-SBA-15 MMS had a perfect hexagonal porous structure (Fig. 3a). And Al species are uniformly distributed on the molecular sieve, both on the framework and outside the framework (Fig. 3d). No large Al species was observed, which indicates EFAL evenly distributed.

The hydrothermal stability of obtained Al-SBA-15 MMS was examined at 100 and 600 °C according to previous studies[13–15]. The structure and specific surface area of Al-SBA-15 samples after hydrothermal aging were characterized by XRD and N$_2$ adsorption–desorption (see Supplementary Figs. 2 and 3, Supplementary Table 2, and Supplementary Note 2), respectively. The results show that all the Al-SBA-15 MMS basically maintained their original morphology after hydrothermal treatment. However, the micropore structure of the Al-SBA-15 molecular sieve disappeared and the mesopore structure was partially destroyed after hydrothermal treatment at 600 °C. Our findings are consistent with previous studies[13–15].

**Properties of Fe–Mn/Al-SBA-15 catalysts.** The NO$_x$ conversion of the Fe–Mn/Al-SBA-15 catalyst was significantly improved at low temperature (Fig. 4a). At 150 °C, the NO$_x$ conversion was increased from 62.8 to 91.1%, and the efficient denitrification temperature window was widened. Figure 4b shows that the N$_2$ selectivity of the catalyst was significantly improved after doping with Al, and the N$_2$ selectivity was higher than 85% at 100–350 °C. From the XPS results shown in Fig. 4c, after Al doping, the valence state of Fe was not changed. However, the valence state of Mn was greatly changed with more Mn$^{3+}$. In combination with Table 1, the proportion of chemisorbed oxygen (O$_\beta$) and lattice oxygen (O$_\alpha$) were also reduced. The XRD patterns shown in Fig. 4d demonstrate that β-MnO$_2$ is the only Mn species on the surface of Fe–Mn/SBA-15. In addition, there is no diffraction peak of Fe$_2$O$_3$, which indicates that Fe$_2$O$_3$ was highly dispersed on the surface of the molecular sieve. In the Fe–Mn/Al-SBA-15 catalyst, a large amount of Mn$_2$O$_3$ was formed, which indicated that the Al doping can induce the transformation of Mn oxide polymorphs. Figure 4e shows the NH$_3$-temperature programmed desorption (TPD) curve for the Fe–Mn/Al-SBA-15

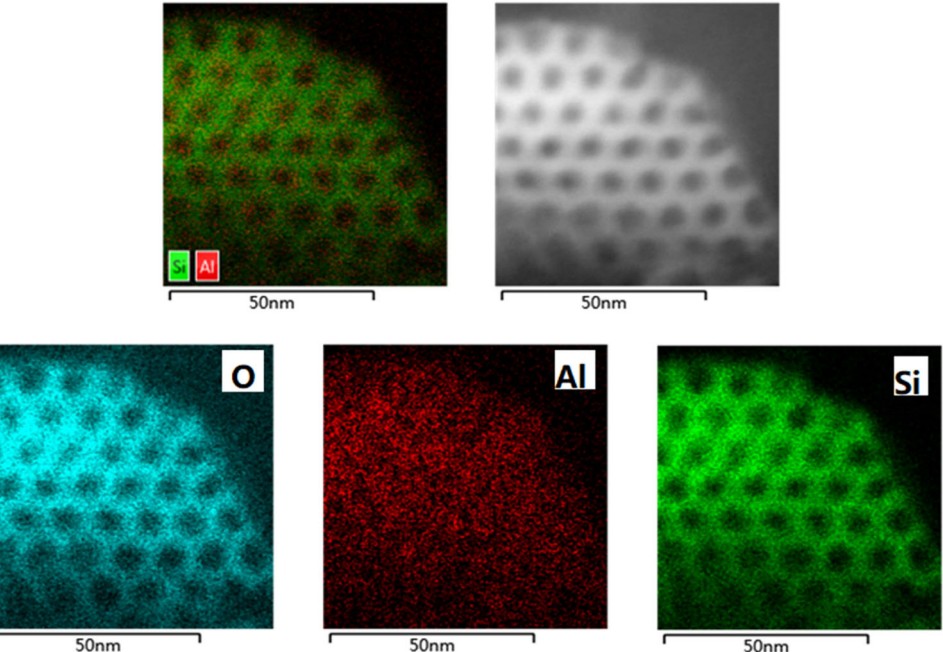

**Fig. 3 HAADF-STEM images of the Al-SBA-15 MMS. a** the Al-SBA-15 sample in the [100] orientation. **b** the image of the same area, showing the contrasts of the different types of metal species. **c** O element distribution images of the same area, which shows the presence of O in the framework. **d** Al element distribution images of the same area, which shows Al species are uniformly distributed on the molecular sieve, both on the framework and outside the framework. **e** Si element distribution images of the same area, which shows the presence of Si in the framework.

catalyst and Fe–Mn/SBA-15 catalyst. The desorption peaks at 100–250, 280–330, and 380–500 °C were attributed to desorption of physisorbed $NH_3$, $NH_3$ bound to weak Brønsted acid sites, and $NH_3$ bound to strong Brønsted and Lewis acid sites, respectively, on the catalyst surface[6]. The total areas of peaks for Fe–Mn/Al-SBA-15 were clearly bigger than that for Fe–Mn/SBA-15. This indicated that $NH_3$ desorption from the Fe–Mn/Al-SBA-15 surface was easier.

Nanometric Fe and Mn species can be identified according to their different contrasts in the HAADF-STEM images. Figure 5 evidences that the active components can highly dispersed both in Fe–Mn/SBA-15 catalyst and Fe–Mn/Al-SBA-15 catalyst. In the Fe–Mn/SBA-15 catalyst, both Fe and Mn species were present in the pores of the molecular sieve with few outside the pores. The oxide particles in the pores were ~2–5 nm in size. As a comparison, the oxide particles outside the pores were much bigger of 60–80 nm in size. As shown in Fig. 5b, Al species were uniformly dispersed on the molecular sieve framework and extra-framework in the Fe–Mn/Al-SBA-15 catalyst. Fe and Mn species also present in the pores of the molecular sieve and outside the pores. However, the number of manganese oxide particles outside the pores is significantly increased with a greatly reduced size of 20–40 nm. This change may arise from the EFAL, which induced the crystal growth of manganese oxide outside the pores.

The redox properties of catalysts can greatly affect the $NH_3$-SCR performance[35–43]. $H_2$-temperature-programmed reduction (TPR) studies were therefore performed on the Fe–Mn/Al-SBA-15 and Fe–Mn/SBA-15 catalysts. The results are shown in Supplementary Fig. 4, Supplementary Table 3, and Supplementary Note 3. The results show that the area of the low-temperature reduction peak for the Fe–Mn/Al-SBA-15 catalyst was relatively large and peaks shift to low temperature. This indicates that Al doping can improve the low-temperature reducibility of the catalyst to benefit the denitrification.

The in situ IR spectra of the Fe–Mn/SBA-15[21] and Fe–Mn/Al-SBA-15 catalyst were measured to understand the origin of their

different catalytic performance. Figure 6a shows that three nitrate peaks at $1630 \, cm^{-1}$ (bridged bidentate nitrates), $1540 \, cm^{-1}$ (monodentate nitrates), and $1505 \, cm^{-1}$ (monodentate nitrates) appeared when 800 ppm NO and 3% $O_2/N_2$ were introduced to Fe–Mn/Al-SBA-15 sample for 10 min at 200 °C. Bidentate nitrates are formed from less stable nitrates and nitrites. The linear nitrite peak at $1489 \, cm^{-1}$, monodentate nitrite peaks at $1406 \, cm^{-1}$ and bidentate nitrite peaks at $1316 \, cm^{-1}$ also appeared[41]. Only two strong peaks appeared at 1630 and $1597 \, cm^{-1}$ were observed in DRIFT of Fe–Mn/SBA-15 sample, which are related to bridged bidentate nitrate and chelate bidentate nitrate, respectively[21]. Those nitrite peaks were not observed in the Fe–Mn/SBA-15 spectrum. This indicated that both the acidity of the catalyst and nitrite adsorption on the catalyst surface increase after Al doping. After further exposure to NO and $O_2$, the intensities of these nitrite peaks gradually decreased. Specific details are available in Supplementary Fig. 5. This shows that nitrite was less stable on the surface of the catalyst, more active and easier to desorb or react compared with nitrate. According to literature reports, nitrite and nitrate are preferably decomposed to $N_2$ and $N_2O$, respectively[38,39]. This is one of the reasons why the $N_2$ selectivity of Fe–Mn/Al-SBA-15 is better than that of Fe–Mn/SBA-15. When 800 ppm $NH_3$ was then passed into the in situ reaction cell, the peaks of the nitrite all disappeared. This shows that the $NH_3$ gas reacts with the nitrite on the catalyst surface. The desorption rate of nitrite increased over $NH_3$ introduction time (Supplementary Fig. 5b).

Figure 6b shows in situ IR spectra of the Fe–Mn/Al-SBA-15 sample under a flow of 800-ppm $NH_3$ at 200 °C. After a 10 min gas flow, peaks from $NH_3$ at Lewis acidic sites also appeared at 1168, and $1254 \, cm^{-1}$, and $NH_4^+$ peaks at Brønsted acidic sites appeared at 1404, 1653, and $1701 \, cm^{-1}$. The vibration peak at $1020 \, cm^{-1}$ corresponds to weakly adsorbed or gas phase $NH_3$ peaks. The peaks from surface-adsorbed $-NH_2$ also appeared at 1509 and $1542 \, cm^{-1}$. With increasing exposure time, the peak intensities of the $NH_3$ species became

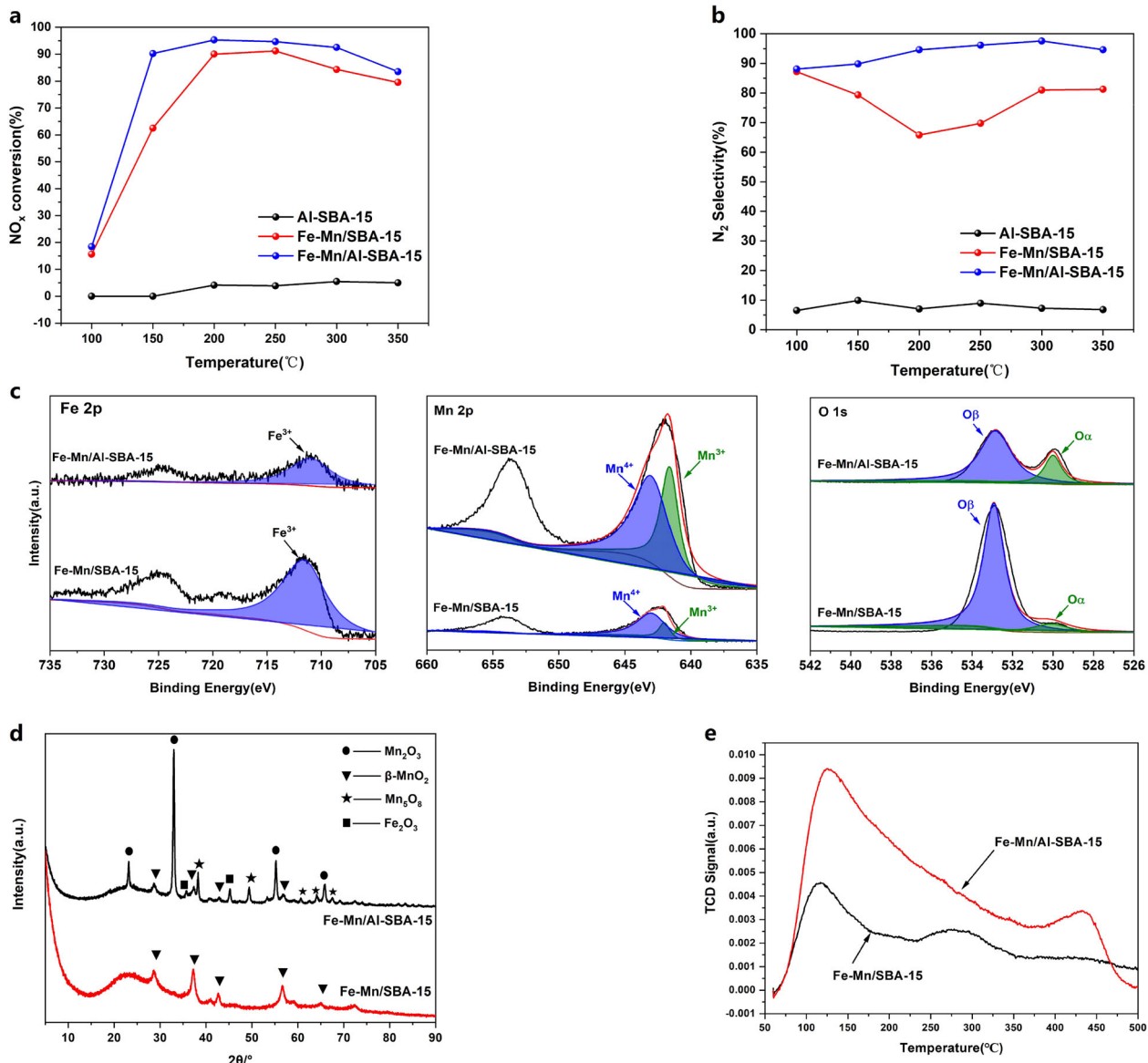

**Fig. 4 NH₃-SCR performances and characterization of Fe–Mn/Al-SBA-15 and Fe–Mn/SBA-15 catalysts. a** NOₓ conversion. **b** N₂ selectivity. **c** XPS spectra. **d** XRD patterns. **e** NH₃-TPD curves.

**Table 1 Binding energies (BEs) and surface atomic concentrations of Mn, Fe, and O in Fe–Mn/Al-SBA-15 catalysts.**

| Samples | Surface concentration (%) | | | Bing energy (eV) | | | | | $Mn^{4+}/Mn^{3+}$ | $O_\beta/O_\alpha$ |
|---|---|---|---|---|---|---|---|---|---|---|
| | Mn2p | Fe2p | O1s | Fe 2p3/2 | O1s | | Mn2p3/2 | | | |
| | | | | | $O_\beta$ | $O_\alpha$ | $Mn^{4+}$ | $Mn^{3+}$ | | |
| Fe–Mn/SBA-15 | 2.42 | 1.38 | 96.20 | 711.41 | 532.92 | 530.09 | 643.26 | 642.07 | 1.88 | 65.10 |
| Fe–Mn/Al-SBA-15 | 16.4 | 0.7 | 82.9 | 711.06 | 532.83 | 530.00 | 643.04 | 642.00 | 1.24 | 2.34 |

increasingly weak, which indicated that these species were not stable on the catalyst surface. The fast desorption of NH₃ species will favor SCR reaction (Supplementary Fig. 5c). In the Fe–Mn/SBA-15 sample, fewer peaks are observed at 1538 cm⁻¹ (surface-adsorbed −NH₂), 1558 cm⁻¹ (NH₃ at Lewis acidic sites) and 1746 cm⁻¹ (NH₄⁺ peaks at Brønsted acidic sites). This supports the NH₃-TPD result that the acid center of Fe–Mn/SBA-15 catalyst is less than that of Fe–Mn/Al-SBA-15 catalyst (Fig. 4e). When 800 ppm NO and 3% O₂ were then passed into the in situ reaction cell, the peaks of nitrate at 1600 and 1629 cm⁻¹ appeared in Fe–Mn/Al-SBA-15 sample, respectively. The peak intensities of these two peaks are higher in the Fe–Mn/SBA-15 catalyst. This also shows that the nitrate formed on the Fe–Mn/ SBA-15 catalyst is more stable than that on the Fe–Mn/Al-SBA-15 catalyst. The absence of nitrite peaks suggests that nitrite could be quickly reacting with adsorbed NH₃ species and desorption immediately after it is formed on the surface of the catalyst[43].

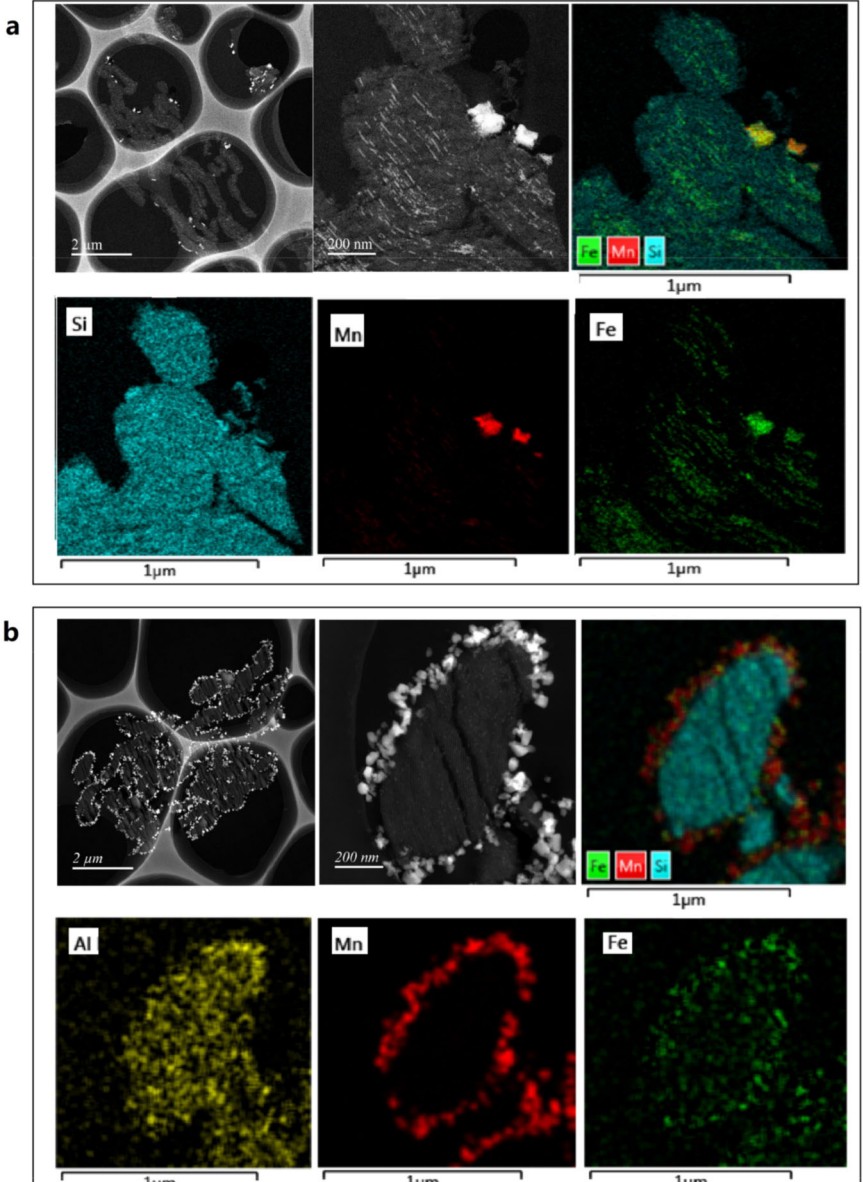

**Fig. 5 HAADF-STEM images of the molecular sieve catalysts. a** Fe–Mn/SBA-15 and **b** Fe–Mn/Al-SBA-15.

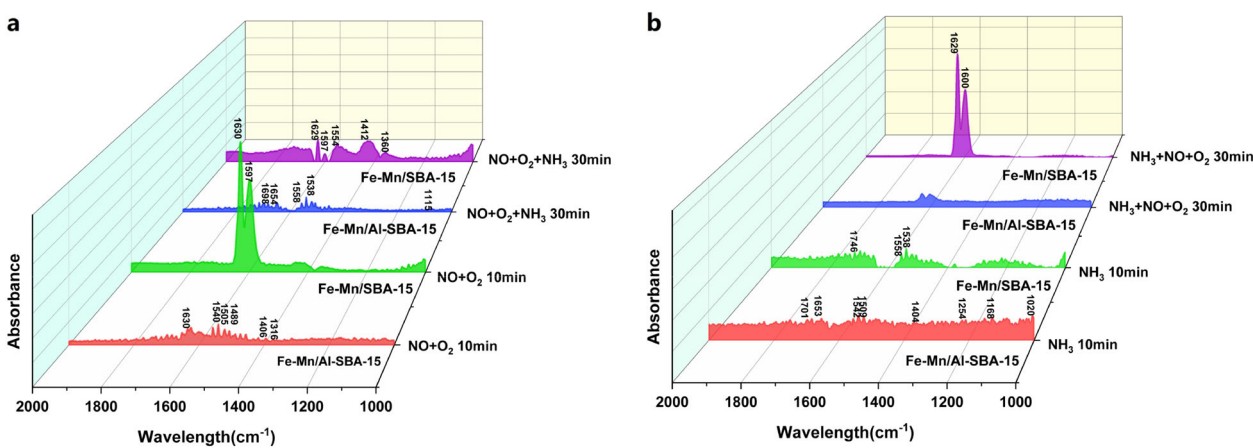

**Fig. 6 DRIFT spectra of Fe–Mn/ SBA-15 and Fe–Mn/ Al-SBA-15 catalysts. a** Exposed to NO and $O_2$ and then treated with $NH_3$ at 200 °C. **b** Exposed to $NH_3$ and then treated with $NO + O_2$ at 200 °C.

All DRIFT spectra of transient (Supplementary Fig. 5e) and steady-state reactions (Supplementary Fig. 5f) on Fe–Mn/Al-SBA-15 catalyst demonstrate that $NH_3$ at Lewis acidic sites and $NH_4^+$ at Brønsted acidic sites had high catalytic activities, and $-NH_2$ and $-NH_4NO_2$ were the intermediate species in the reaction. The transient reaction of $NO + O_2$ resulted in the adsorption of nitrite and weakly adsorbed species in the gas phase as NO active species. The SCR reaction, therefore, mainly occurred through the two adsorbed states of $NH_3$ and nitrite, and weakly adsorbed NO in the gas phase. The Langmuir–Hinshelwood and Eley–Rideal mechanisms may be simultaneously involved. All DRIFT spectra of Fe–Mn/SBA-15 catalyst are shown in Supplementary Fig. 6. The possible SCR reaction routes on Fe–Mn/Al-SBA-15 catalyst are illustrated in Supplementary Fig. 7.

Under practical application conditions, the exhaust gases contain small amounts of $SO_2$ and $H_2O$ that can poison catalyst. This is because $SO_2$ can be catalytically oxidized to $SO_3$ by the catalyst, then $SO_3$ can react with catalysts to irreversibly form metal sulfates or ammonium sulfates, that is, $NH_4HSO_4$ and $(NH_4)_2SO_4$. These ammonium sulfates can physically block the catalyst and most of the active sites, which results in the declined $NO_x$ conversion. Therefore, the influence of $SO_2$ and $H_2O$ on the catalytic performance was investigated. The results shown in Supplementary Fig. 8 suggest that the $NO_x$ conversion decreased to 64.79% in 1 h in the presence of 500 ppm of $SO_2$ and 5% $H_2O$. The catalyst slightly recovered after stopping the feed of $H_2O$ and $SO_2$. However, the $NO_x$ conversion only maintained at 65–70%. Although the denitrification efficiency decreases intermediately after the treatment of $H_2O$ and $SO_2$, it remains stable over the time. In industrial applications, the reduced airspeed may increase the denitrification efficiency.

The low-temperature $NH_3$-SCR performance of reported $deNO_x$ catalysts is summarized in Supplementary Table 4. Supplementary Table 4 and Supplementary Note 4 show similar denitrification efficiency compared with the reported low-temperature denitrification catalysts. However, we can achieve the same performance using faster gas hourly space velocity (GHSV) and lower gas concentrations in comparison with previous studies. Moreover, the developed catalysts of this work are considerably less costly and free of V. They are, thus, more suitable for industrial applications.

**Density functional theory (DFT) calculations**. To understand the role of Al in the catalytic process, DFT calculations were carried out[44–50]. The XRD data revealed that the introduction of Al caused the Mn-based species to change from $MnO_2$ to $Mn_2O_3$.

To validate the role of Al, the phase change energies were calculated based on the following reactions:

$$2Mn_2O_3 + O_2 = 4MnO_2, \tag{2}$$

$$2Al_{0.125}Mn_{1.875}O_3 + O_2 = 4Al_{0.0625}Mn_{0.9375}O_2, \tag{3}$$

$$2Si_{0.125}Mn_{1.875}O_3 + O_2 = 4Si_{0.0625}Mn_{0.9375}O_2. \tag{4}$$

The reaction energies for these three reactions were 0.040, 0.174, and −0.046 eV, respectively. Here, the negative values suggest that the reaction is thermodynamically allowed. The reaction energy of the first reaction suggested that $Mn_2O_3$ is slightly more stable than $MnO_2$. Even after the doping of a low concentration of Al of 6.25%, the reaction energy was more than four times larger than that without any dopant, which indicates that the Al can further stabilize the $Mn_2O_3$ polymorph. However, the $MnO_2$ became more stable when 6.25% Si dopants are introduced. The thermodynamic analyses explain why the $MnO_2$ became the dominant species when a SBA-15 support only with Si cations is employed and the $Mn_2O_3$ phase is the main Mn-based species after Al is introduced in the support as evidenced by the XRD image (Fig. 4d).

The $N_2$ selectivity of $NH_3$-SCR catalyst was directly related to the interaction strength of the reactants and intermediates with the surface. DFT calculations on the adsorption properties of relevant small molecules on $MnO_2$ (110) and $Mn_2O_3$ (222) surfaces were performed. These two surfaces were selected because they may be the most exposed facets from the XRD result. Only the Mn species were focused for this comparative study because previous experimental results suggest that they play the decisive role for the $N_2$ selectivity in the SCR reaction[21]. The adsorption properties of NO and $NH_3$ were calculated, since both are the reactants. The configuration of adsorbed $NO_2$ with an adjacent O vacancy was also considered because it was one of the important intermediates during the SCR reaction. Indeed, the total energies of the configuration of $NO_2$ with an O vacancy on both surfaces are lower than that with adsorbed NO on the clean surfaces, which suggested that the adsorbed NO could easily be oxidized by the crystal oxygen. After the formation of $NO_2$, the adsorption energy of $NO_2$ on the defective surface with the oxygen vacancy is close to zero, which indicates that the $NO_2$ on the surface is considerably mobile to form $N_2O_4$ for further reduction, as illustrated in Fig. 7.

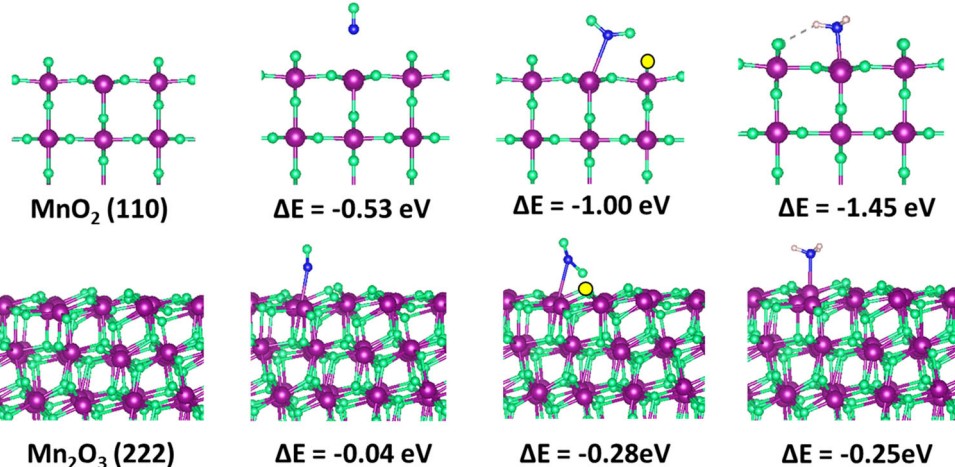

**Fig. 7 The adsorption of NO and $NH_3$ on $MnO_2$ (110) and $Mn_2O_3$ (222) surfaces.** Color code: purple: Mn, green: O, yellow: O vacancy, blue: N, and gray: H.

## Discussion

Fly ash-derived Al-SBA-15 MMS with different Si:Al molar ratios were prepared by a post-synthesis impregnation method without water. The effects of pH value and the amount of Al added on the structure of the molecular sieve were investigated. The results show that Al is successfully doped in SBA-15 MMS. The acidity (acid position, acidic strength, L acid/B acid ratio, and L acid and B acid synergy) were investigated by in situ FTIR with the transmission mode, Py-IR, TMP-$^{31}$P MAS NMR, $^{13}$C acetone-$^{1}$H/$^{13}$C CP MAS, $^{1}$H/$^{27}$Al TRAPDOR NMR, and two-dimensional $^{1}$H-$^{1}$H DQ-MAS NMR. All characterization results confirm that Al exists not only in the framework but also in the extra-framework, which correspondingly provided B acid and L acid to the catalyst. The strength of L acid was greater than the strength of B acid. The spatial proximity of L acid and B acid enhanced their synergistic effect and greatly enhanced their acidity.

The denitrification activities of a series of SCR catalysts were investigated. At 150–300 °C, the denitrification efficiency of the Fe–Mn/Al-SBA-15 catalyst was higher than 90%. Moreover, the N$_2$ selectivity of the Fe–Mn/Al-SBA-15 catalyst remained above 86% at 150–300 °C, which was higher than that of the Fe–Mn/SBA-15 catalyst. This was the result of a combination of acidity, redox properties, and synergistic effects in the molecular sieve catalysts. The XRD, XPS, HAADF-STEM, and DFT results show that Al dopants can induce the growth of Mn$_2$O$_3$ catalysts, which is beneficial for NH$_3$-SCR. The denitrification reaction mechanism for the Fe–Mn/Al-SBA-15 catalyst at 200 °C was investigated by in situ IR spectroscopy. The results showed that many nitrite peaks were present after doping with Al, which further indicated that Al doping increased the catalyst acidity and improved nitrite adsorption on the catalyst surface. The E–R and L–H mechanisms were simultaneously involved, and the E–R mechanism dominated.

Based on the DFT results, there is a stronger interaction between NO and MnO$_2$, which indicates that Fe–Mn/SBA-15 possesses a strong reactivity for NO conversion. This matches the reported experimental data[21]. However, the adsorption strength of NH$_3$ on the MnO$_2$ (110) surface is too strong, which limits its reduction capability. As a comparison, the adsorption energy of NH$_3$ on Mn$_2$O$_3$ (222) was −0.25 eV, which benefits the desorption of NH$_3$ to selective produce N$_2$. The DFT data also confirm the DRIFT analysis conclusion that NH$_3$ is unstable on the Fe–Mn/Al-SBA-15 catalyst with respect to that on the Fe–Mn/SBA-15 catalyst. The desorption of NH$_3$ is a crucial step in the SCR reaction for the selectively produce N$_2$. This explains the high N$_2$ selectivity when Mn$_2$O$_3$ acts as the active site in a catalyst. Thus, the Al-induced component engineering for the formation of Mn$_2$O$_3$ is essential for the advance of low-temperature NH$_3$-SCR.

## Methods

**Synthesis of fly ash-derived SBA-15 MMS and AlCl$_3$ powders**. The chemical composition and powder XRD pattern of high-alumina fly ash are shown in Supplementary Method (Supplementary Fig. 9 and Supplementary Note 5). Hydrochloric acid was used to extract alumina in the fly ash. The Al leaching rates from fly ash were investigated for various grinding times and acid concentrations, as shown in Supplementary Fig. 10, along with a detailed explanation in Supplementary Notes 6 and 7. The main chemical components of the extracted Al solution were Al$_2$O$_3$ 357.41 g/L, Fe$_2$O$_3$ 7.69 g/L, CaO 10.75 g/L, MgO 0.95 g/L, and SiO$_2$ 0.178 g/L. The filter cake obtained by extracting Al from the fly ash was Si rich. Its main chemical components were 80.51% SiO$_2$, 13.34% Al$_2$O$_3$, 0.25% CaO, and 0.32% Fe$_2$O$_3$. The main phases were mullite, quartz, and anatase. The chemical composition of the Si-rich filter cake was similar to that of the fly ash; therefore, we used the experimental conditions previously used for Si extraction by a fly ash alkali dissolution method for Si-leaching experiments. Details are reported in our previous publications[19,20].

The SBA-15 MMS synthesized hydrothermally at 100 °C had a three-dimensional porous structure (Supplementary Figs. 11 and 12, Supplementary Notes 8 and 9). Main pore channels were produced and disordered microporous wall and mesoporous tunnels were obtained. The BET surface area of the prepared fly ash-derived SBA-15 MMS was 793.59 m$^2$/g, the pore volume was 0.748 cm$^3$/g, and the average pore diameter was 6.11 nm. The main chemical components of the SBA-15 MMS were SiO$_2$ 98.81%, Na$_2$O 0.53%, Fe$_2$O$_3$ 0.006%, P$_2$O$_5$ 0.003%, Cl 0.40%, K$_2$O 0.004%, SO$_3$ 0.22%, and others 0.03%.

The main chemical components of the resulting AlCl$_3$ crystals were Al$_2$O$_3$ 97.41%, Fe$_2$O$_3$ 0.69%, CaO 0.005%, MgO 0.0015%, and Cl 22.6%. Supplementary Fig. 13 and Supplementary Note 10 show that the prepared AlCl$_3$ crystal was mainly composed of AlCl$_3$·6H$_2$O, and no other impurity peaks appeared, which indicated that the prepared AlCl$_3$ crystals were relatively pure.

**Synthesis of fly ash-derived Al-SBA-15 MMS**. A series of Al-SBA-15 catalysts were prepared through a impregnation method with prepared AlCl$_3$ crystals as the Al precursors and the resulting fly ash-derived SBA-15 MMS as the support; various molar ratios of Si:Al were used. A certain quality of AlCl$_3$ crystals and solvent (H$_2$O/absolute ethanol) were mixed with SBA-15 (1 g) and the mixture was stirred for 24 h at 60 °C. After, the mixture was dried at 100 °C for 10 h, and then calcined in air at 550 °C for 5 h. The heating rate was 5 °C/min. Al-SBA-15 MMS with various Si:Al molar ratios were obtained. It should be noted that the Si/Al ratios were calculated from chemical stoichiometric compositions used in the sample syntheses.

**Synthesis of Fe-Mn/Al-SBA-15 MMS catalyst**. An Fe–Mn/Al-SBA-15 MMS catalyst was prepared by the same method. Al-SBA-15 (1 g) was added to Mn(NO$_3$)$_2$ and Fe(NO$_3$)$_2$ solutions at a Mn:Fe molar ratio of 1:1. The resulting materials were denoted by $m$Fe-$n$Mn/Al-SBA-15, where $m$ and $n$ represent the Fe and Mn weight percentage loadings, respectively.

The fly ash-derived Fe–Mn/Al-SBA-15 preparation route is shown in Fig. 8. Fe–Mn/SBA-15 materials were also prepared for comparison.

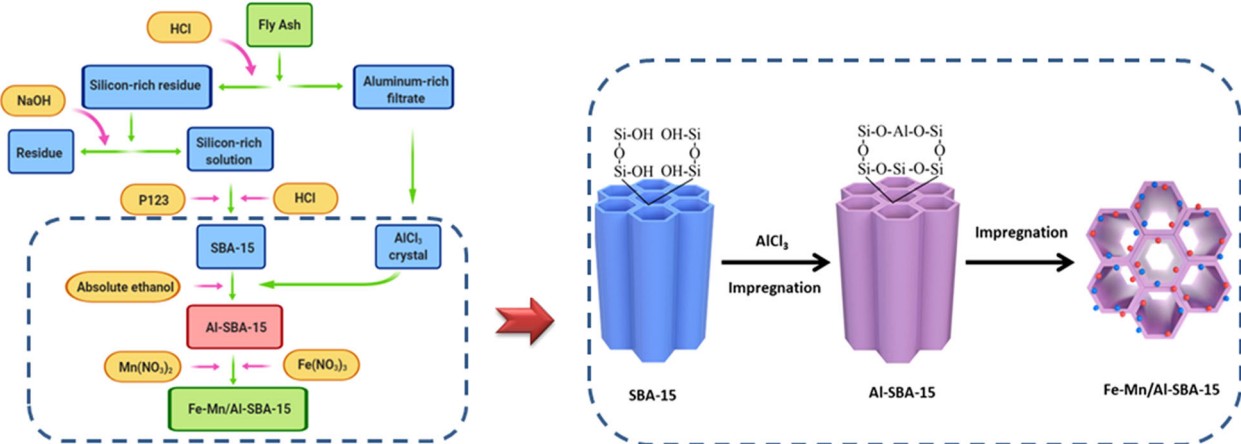

**Fig. 8 Preparation of fly ash-derived Fe–Mn/Al-SBA-15.** Schematic illustration for preparation process of Fe–Mn/Al-SBA-15 catalyst from fly ash. The dotted box on the right is a schematic diagram of the dotted box on the left.

**Characterizations**. XRD was performed with a D8 ADVANCE diffractometer (Bruker, Germany), using Cu Kα radiation and a step size of 0.02°. Small-angle data were collected from 0.5° to 5° at a scanning rate of 0.5°/min. Wide-angle data were collected from 5° to 90° at a scanning rate of 3°/min.

N$_2$ adsorption–desorption isotherms were recorded using an ASAP2020 automatic physical adsorption instrument (Micromeritics, USA) at 77 K after degassing the samples at 373 K.

TEM analyses were performed on a JEOL 200F TEM/STEM microscope operating at 200 kV. The HAADF images and elemental mapping were taken on a JEM-200F TEM (JEOL Co. Ltd) equipped with a Cs probe corrector (CEOS, Gmbh) and a 100 mm$^2$ active area EDS detector (X-Max$^N$100TLE, Oxford Instruments plc). Before observation, the SBA-15 powders were embedded in Epoxy resin, cut to 50 nm thin slices by ultramicrotome (EM UC7, Leica Microsystems Inc), and transferred onto holey carbon film.

XPS analysis was performed on an ESCALAB 250xi (Thermo Scientific, UK) using monochromatic Al Kα radiation (1486.6 eV) operating at 25 W.

H$_2$-TPR experiments were performed with an AutoChem II 2920 instrument (USA). The catalyst (0.3 g) was pretreated in an Ar flow (50 mL/min) for 30 min at 500 °C to remove water and other impurities. As the samples cooled, the Ar flow was replaced by a reductive mixture of 10.0% H$_2$ in Ar and the reactor temperature was raised to 800 °C at a heating rate of 10 °C/min.

NH$_3$-TPD experiments were performed with an automatic physical and chemical adsorption instrument (AutoChem II 2920, Micromeritics). Before adsorption of NH$_3$ at 373 K, the samples were heated at 773 K in a He flow. The amount of NH$_3$ desorbed between 373 and 873 K at a heating rate of 10 K/min was determined with an on-line gas chromatograph equipped with a thermal conductivity detector.

$^{27}$Al and $^{29}$Si MAS NMR spectra were recorded with a Bruker Avance III HD/ 89 mm instrument. Tetraethyl orthosilicate and Al(H$_2$O)$_6$$^{3+}$ were used as the references for $^{29}$Si and $^{27}$Al, respectively.

Prior to solid-state NMR experiments, the Al-SBA-15 samples were placed in glass tubes and dehydrated at 673 K under a pressure below 2.0 Pa for 10 h on a vacuum line. After the dehydrated samples were cooled to room temperature, 2.4 molecule/u.c. of 2-$^{13}$C-acetone was introduced and frozen by liquid N$_2$. Saturated adsorption of TMP onto the Al-SBA-15 samples was carried out in a similar way. Finally, the samples upon adsorption of the probe molecule as well as the dehydrated Al-SBA-15 samples were flame sealed. The sealed samples were transferred into a ZrO$_2$ rotor (tightly sealed by a Kel-F cap) under a dry nitrogen atmosphere in a glove box.

Solid-state NMR experiments were carried out on a Bruker Avance III 500 spectrometer with a 4-mm triple-resonance MAS probe. The Larmor frequencies were 500.6, 125.9, and 202.6 MHz for $^1$H, $^{13}$C and $^{31}$P nuclei, respectively. $^1$H spin echo MAS NMR spectra were acquired with a π/2 pulse length of 5.0 μs and a recycle delay of 2 s. The $^1$H/$^{27}$Al TRAPDOR experiment was carried out with a spinning speed of 14 kHz, an irradiation time of 143 μs, and a radio frequency field strength of 50 kHz for $^{27}$Al. In the 2D $^1$H-$^1$H DQ-SQ MAS NMR experiments, double-quantum coherences of 286 μs and a spinning speed of 14 kHz were applied. The increment interval in the indirect dimension was set to 71.4 μs. 128 t1 increments and 96 scan accumulations for each t1 increment were used. $^{13}$C CP/MAS NMR experiments were performed with a contact time of 1.2 ms, a recycle delay of 2 s, number of scans of 6000, and MAS of 14 kHz. $^{31}$P CP/MAS NMR experiments were performed with a contact time of 1.5 ms, a recycle delay of 2 s, number of scans of 1000, and MAS of 10 kHz. The chemical shifts of $^1$H and $^{13}$C were externally referenced to adamantine, whereas the $^{31}$P chemical shifts were calibrated using (NH$_4$)$_2$HPO$_4$.

In situ transmission IR sample activation conditions: the sample was a self-supporting piece with a diameter of 13 mm. The temperature was increased at a rate of 10 °C/min from room temperature to 500 °C, held for 60 min, rapidly reduced to 350 °C, the kept stable for 5 min sampling (resolution 4 cm$^{-1}$, number of scans 64).

DRIFT spectra were recorded with an IR Prestige-21 instrument (Shimadzu) at a resolution of 4 cm$^{-1}$ and averaged over 500 scans. These studies were performed by heating pre-calcined powder samples in situ from room temperature to 673 K at a heating rate of 5 K/min in a pure N$_2$ flow (40 mL/min). The samples were kept at 673 K for 3 h and then cooled to 323 K. Py vapor (20 μL) was then introduced under a N$_2$ flow; IR spectra were recorded at various stages of Py desorption, which was maintained by evacuation at progressively higher temperatures (323–473 K). A resolution of 4 cm$^{-1}$ was attained after averaging 500 scans for all the IR spectra recorded. XPS was performed with an ESCALAB 250xi instrument (Thermo Scientific, UK), using monochromatic Al Kα radiation (1486.6 eV), at 25 W. The sample was outgassed overnight at room temperature in an ultrahigh-vacuum chamber (<5 × 10$^{-7}$ Pa). All binding energies were referenced to the C 1 s peak at 284.6 eV. The experimental errors were within ±0.1 eV.

**SCR activity measurements**. The SCR reaction was evaluated in a fixed-bed reactor. The sample (0.3 g) was put in a reaction tube and the tube was placed in simulated flue gas for 2 h. The gas mixture contained 300 ppm NO, 300 ppm NH$_3$, and 3% O$_2$, with N$_2$ as the balancing gas; the GHSV was ~120,000 h$^{-1}$. The catalytic activity was determined by analyzing the inlet and outlet gases with a flue gas analyzer (MultiGas™ 6030, MKS) at temperatures between 100 and 300 °C. The

NO$_x$ conversion and N$_2$ selectivity were calculated as follows:

$$NO_x \text{ conversion} = \left[ \left( [NO_x]_{in} - [NO_x]_{out} \right) / [NO_x]_{in} \right] \times 100\%, \quad (5)$$

$$N_2 \text{ selectivity} = \left( 1 - \frac{2[N_2O]_{out}}{[NO_x]_{in} + [NH3]_{in} - [NO_x]_{out} - [NH3]_{out}} \right) \times 100\%, \quad (6)$$

where NO$_x$ is the sum of the NO and NO$_2$ concentrations, [NO$_x$]$_{out}$ is the outlet concentration of NO$_x$, [NO$_x$]$_{in}$ is the inlet concentration of NO$_x$, [NH$_3$]$_{out}$ is the outlet concentration of NH$_3$, [NH$_3$]$_{in}$ is the inlet concentration of NH$_3$, and [N$_2$O]$_{out}$ is the outlet concentration of N$_2$O.

**DFT calculations**. The spin-polarization DFT computations were conducted using the Vienna ab initio simulation package based on the projector augmented wave (PAW) method[48–50]. Electron–ion interactions were described using standard PAW potentials, with valence configurations of 3p$^6$4s$^2$3d$^5$ for Mn, 2s$^2$2p$^4$ for O (O), 2s$^2$2p$^3$ for N, and 1s$^1$ for H. A plane-wave basis set was employed to expand the smooth part of wave functions with a cut-off kinetic energy of 520 eV. The exchange and correlation functional parameterized by Perdew–Burke–Ernzerhof[48–50], a form of the general gradient approximation, was used throughout. Owing to the large Coulombic repulsion between localized d electrons of transition metals, the DFT+U method was used to correct the material properties of transition metal oxides, particularly for magnetic ground states and electronic structures. For Mn, the U–J value was set as 3.5 eV.

The MnO$_2$(110) and Mn$_2$O$_3$ (222) were simulated using the 12-layer slab model with the consideration of their magnetic structures. A sufficiently large vacuum region of 15 Å was used to ensure that the periodic images were well separated. The bottom six layers were fixed at the bulk position. The other atoms were allowed to relax during the structural optimization. The (2 × 3) and (1 × 1) surface cells were employed for MnO$_2$(110) and Mn$_2$O$_3$ (222), respectively, for the study on the adsorption of NO and NH$_3$ with one adsorbate molecule on the topmost surface layer. The corresponding k-point grids were gamma-centered (4 × 4 × 1) and (2 × 2 × 1) for MnO$_2$(110) and Mn$_2$O$_3$ (222), respectively. The properties of NO, NO$_2$, and NH$_3$ molecules were calculated in a 20 × 20 × 20 Å$^3$ box. The convergence criterion for the electronic self-consistent loop was set to 10$^{-5}$ eV. The atomic structures were optimized until the residual forces were below 0.002 eV Å$^{-1}$. The adsorption energy can be defined as follows:

$$\Delta E_{ads} = E_{gas/substrate} - E_{substrate} - E_{gas}, \quad (7)$$

where $E_{gas/substrate}$ is the total electronic energy of the substrate with an adsorbed gas molecule and $E_{substrate}$ and $E_{gas}$ correspond to the energy of the substrate and the gas molecule, respectively, in a vacuum.

## Data availability

The data supporting the findings of this study are available within the article and its Supplementary Information files, or from the corresponding authors on reasonable request.

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

## Acknowledgements

We gratefully acknowledge S.H. Li for help in NMR measurement and data analysis. The NMR experiment was performed at the Wuhan Institute of Physics and Mathematics, Chinese Academy of Science. We thank S.S. Liu for help with the in situ DRIFTS measurements. The in situ DRIFTS experiment was performed at the College of Chemical Engineering, Beijing University of Chemical Technology. This work was supported by the National High Technology Research and Development Program ("863" program) of China (2012AA06A115), National Key Research and Development Program of China (2019YFC1907500), and China Postdoctoral Science Foundation (2017M610723). This research was undertaken on the supercomputers at the National Computational Infrastructure (NCI) in Canberra, Australia, which is supported by the Australian Commonwealth Government, and Pawsey Supercomputing Center in Perth with funding from the Australian Government and the Government of Western Australia.

## Author contributions

G.L. carried out all relevant experiments, data, analysis and wrote the first draft of the paper. B.W., Q.S., and Y.W. contributed to the data analysis and paper writing. B.W. and Q.S. supervised the experiments. Y.W. performed all DFT calculation. Z.H. did HAADF-STEM measurements. C.Z. drew some figures. Z.M., H.W., and J.Z. carried out some NH₃-SCR measurements. J.M., D.L., and F.H. conducted some catalyst characterizations.

## Competing interests

The authors declare no competing interests.
