## [Peer Review File · Communications Chemistry]

Reviewers' comments:

Reviewer #1 (Remarks to the Author):

Emission control is one of the toughest tasks that Chinese society is facing. However, the intensive dependency of coal as the preliminary energy makes the NO_x emission control very challenging and costly. This research uses one solid waste fly ash obtained from coal-fired power plant as raw material to prepare molecular sieve denitrification catalyst for NO_x treatment. This technology simultaneously solves the problem of two major pollutants in coal-fired power plants. The results will enable the development of new methods for fly ash utilization, and provide technical information which will help in controlling nitrogen oxide emissions in flue gases. The nature of acid sites, acid strength and Bronsted/Lewis acid synergy in Al-SBA-15 materials was studied by various NMR techniques including ²⁷Al, ²⁹Si, ¹H ultra-high speed MAS (magic-angle spinning), TMP-³¹P MAS, acetone-¹H/¹³C CP MAS, ¹H-¹H DQ-MAS and double-resonance methods such as ¹H/²⁷Al transfer of population in double resonance (TRAPDOR). This article has theoretical guidance for the introduction of acidity in mesoporous molecular sieves. It also points out how mesoporous molecular sieves can be used in the field of catalysis. The acidity induced active component of mesoporous molecular sieves for low-temperature NH₃-SCR was discovered. The DFT simulation calculation, XRD, XPS, HAADF-STEM, in-situ IR spectra and other experimental results reveal that the introduction of Al induces the growth of Mn₂O₃ catalysts. This finding, provide a novel strategy for the rational design of low-temperature NH₃-SCR catalysis through the dopant-induced component engineering of composite catalysts. In all, I think this work is high of novelty and can be accepted for publication after some minor revisions:

1. It would be helpful if the authors would demonstrate how the denitrification efficiency at some selected low-temperatures is compared to few of the state-of-the-art low-T NH₃-SCR catalysts reported so far. It is important of course that the authors should choose similar experimental conditions as those used in their work (feed composition, GHSV).
2. In many previous researches, manganese-based catalyst has weak sulfur resistance. The authors should consider the activity performance in the presence of H₂O and SO₂ in the feed stream and its long-term stability.
3. As we all know, the hydrothermal stability of silica-aluminolite molecular sieve is not good. Has the author tested the hydrothermal stability of Al-SBA-15 molecular sieve?

Reviewer #2 (Remarks to the Author):

This manuscript reports on the development of novel low temperature catalysts for selective NH₃ catalytic reduction of NO_x pollutants. Conventional commercial catalysts do not work at flue temperatures below 300 °C, which heightens the importance of this study. Furthermore, the large process scale of reducing stationary NO_x emissions in power plants highlights the technological and economical significance of the work.

The study builds on previous work from the authors and focuses on the efficient doping of Al in SBA-15 mesoporous molecular sieves. Introduction of Al could induces the formation of Mn₂O₃ that enhances the selectivity to N₂. The authors rightly identify the importance of the acidity of the SBA-15 support and study the effects of pH as well as Si/Al ratio before they carry on to develop the optimal Fe-Mn/Al-SBA-15 catalyst. The authors have given a lot of thought in preparation of their catalyst development strategy, they used advanced techniques for catalyst sample characterisation, such as XRD, XPS, IR, TEM, NMR and TPD, as well as a continuous reactor system where they demonstrate the superior activity and selectivity of the developed catalyst. Although the study is mainly experimental, it is complemented by preliminary DFT theoretical calculations too.

Besides being well designed, the experimental work is carefully carried out and the results critically interpreted. The arguments of the authors are well presented, easy to follow and supported by

literature references. The manuscript is well written with very important results and should be accepted as it is.

Reviewer #3 (Remarks to the Author):

In this manuscript, the authors reported a novel catalytic system of Fe-Mn/Al-SBA-15 to reduce nitrogen monoxide with ammonia. The author discussed that Al in fly ash can guide the growth of Mn₂O₃ in the catalyst. This study sheds new light on the design and fabrication of novel catalytic system to solve the environmental crisis. However, to make this manuscript stronger in terms of scientific importance, I believe this research article could be accepted for publication after revision. The detailed comments are as followings:

- (1) As we all know, the water resistance and sulfur resistance of the catalysts are also important in the reaction, please supplement the experiments.
- (2) Two different acid sites were described in the particle. Please explain briefly which acid site played a key role in this system. Some references about the acidic sites can increase catalytic activity should be cited and compared, such as *Angew. Chem.* (2019, 58, 6351); *Environ. Sci. Technol.* (2019, 53, 938–945) and others.
- (3) In figure 4, whether the peaks of the known elements were optimally fitted, please explain briefly. And please add the title of the ordinate in the paper.
- (4) Grammar and expression should be reconsidered in some places.

Response Letter

Reviewer #1:

Emission control is one of the toughest tasks that Chinese society is facing. However, the intensive dependency of coal as the preliminary energy makes the NO_x emission control very challenging and costly. This research uses one solid waste fly ash obtained from coal-fired power plant as raw material to prepare molecular sieve denitrification catalyst for NO_x treatment. This technology simultaneously solves the problem of two major pollutants in coal-fired power plants. The results will enable the development of new methods for fly ash utilization, and provide technical information which will help in controlling nitrogen oxide emissions in flue gases. The nature of acid sites, acid strength and Bronsted/Lewis acid synergy in Al-SBA-15 materials was studied by various NMR techniques including ²⁷Al, ²⁹Si, ¹H ultra-high speed MAS (magic-angle spinning), TMP-31P MAS, acetone-¹H/¹³C CP MAS, ¹H-¹H DQ-MAS and double-resonance methods such as ¹H/²⁷Al transfer of population in double resonance (TRAPDOR). This article has theoretical guidance for the introduction of acidity in mesoporous molecular sieves. It also points out how mesoporous molecular sieves can be used in the field of catalysis. The acidity induced active component of mesoporous molecular sieves for low-temperature NH₃-SCR was discovered. The DFT simulation calculation, XRD, XPS, HAADF-STEM, in-situ IR spectra and other experimental results reveal that the introduction of Al induces the growth of Mn₂O₃ catalysts. This finding, provide a novel strategy for the rational design of low-temperature NH₃-SCR catalysis through the dopant-induced component engineering of composite catalysts. In all, I think this work is high of novelty and can be accepted for publication after some minor revisions:

We thank the reviewer for the useful comments and suggestions. Now we have carefully addressed the comments by correcting some sentences to support our results. Our point-to-point responses to reviewer's comments are listed below.

1) It would be helpful if the authors would demonstrate how the denitrification efficiency at some selected low-temperatures is compared to few of the state-of-the-art low-T NH₃-SCR catalysts reported so far. It is important of course that the authors should choose similar experimental conditions as those used in their work (feed composition, GHSV).

Response: *Thanks for the reviewer's comment. A new table is added as Supplementary Table 3 in Supplemental Information to compare low-temperature NH₃-SCR efficiency of different state-of-the-art deNO_x catalysts.*

Supplementary Table 3 Comparison of low-T NH ₃ -SCR efficiency with different deNO _x catalysts.				
Catalyst	Feed Composition	GHSV	NO Conversion/%	Ref.
15Fe-Mn/ZSM-5	600 ppm NO, 600 ppm NH ₃ , 5.0 % O ₂ , 100 ppm SO ₂	45,000 h ⁻¹	90% (100-250°C)	9
12.5Ce-FeMnO _x	0.1% NO, 0.1% NH ₃ , 3% O ₂	30,000 h ⁻¹	95% (100-140°C)	10
FeMnTiO _x	800 ppm NO, 800 ppm NH ₃ , 5.0 % O ₂	30,000 h ⁻¹	100% (100-350°C)	11
Fe _{0.3} Ho _{0.1} Mn _{0.4} /TiO ₂	0.08 % NO, 0.08 % NH ₃ , 5% O ₂	20,000 h ⁻¹	90% (120-200°C)	12
K- α-MnO ₂	500ppm NO, 500ppm NH ₃ , 5%O ₂ , 10%H ₂ O,100ppm SO ₂	60,000h ⁻¹	100% (150-250°C)	13
MnO _x -CeO ₂ nanosphere	500 ppm NO, 500 ppm NH ₃ , 5% O ₂	60,000 h ⁻¹	100% (125-250°C)	14
MnO _x -TiO ₂ NS	1000ppm NO, 1100 ppm NH ₃ , 4% O ₂	50,000 h ⁻¹	80% (200-300°C)	15
Cu-Fe-Ti Co-Fe-Ti	500 ppm NO,500 ppm NH ₃ , 3.5 % O ₂	60,000 h ⁻¹	90% (200-250°C)	16
Fe-Mn/SBA-15	300ppmNO,300ppm NH ₃ ,3%O ₂	120,000h ⁻¹	90% (200-250°C)	17
Fe-Mn/Al-SBA-15	300ppmNO,300ppm NH ₃ ,3%O ₂	120,000h ⁻¹	90% (150-300°C)	This study

Our Fe-Mn/Al-SBA-15 shows similar denitrification efficiency compared with the reported low-temperature denitrification catalysts. However, we can achieve the same performance using faster gas hourly space velocity (GHSV) and lower gas concentrations in comparison with previous studies. According to the E-R and L-H mechanisms, the higher initial gas concentration in the flue gas can lead to the more gaseous reactants actively adsorbed on the surface of the catalyst, which can promote the removal of NO and further increase the denitrification efficiency. In previous studies, the GHSV was relatively low of 20,000h⁻¹ to 60,000 h⁻¹ since the lower GHSV can improve the denitrification efficiency. This is because of the increase of GHSV means the rate of gas molecules passing through the interface increases. As a result, the residence time of gas molecules on active surface or active center of per unit volume in catalyst is shortened. This causes the smaller average adsorption rate of reactants, which reduces the denitrification efficiency. However, we can achieve the same catalytic performance with faster GHSV (120,000 h⁻¹) and lower gas concentrations here. It demonstrates that our obtained Fe-Mn/Al-SBA-15 catalyst has excellent low-temperature catalytic denitrification performance. The relevant discussion has also been added in Supplemental Information.

2) In many previous researches, manganese-based catalyst has weak sulfur resistance. The authors should consider the activity performance in the presence of H₂O and SO₂ in the feed stream and its long-term stability.

Response: *Many thanks for the helpful suggestions. We agree with the reviewer that the study of the sulfur resistance and long-term stability of the catalyst is important. As suggested by the reviewer, the sulfur resistance results has been provided as shown in Supplementary Figure 8 in Supplemental Information. The relevant discussion is supplemented on page 17 in main text.*

Supplementary Fig. 8 The effect of water and/or SO₂ on/off presence in the feed stream and the long-term stability of the Fe-Mn/Al-SBA-15 catalyst at 200 °C. Reaction conditions: [NO]=[NH₃]=300 ppm; [SO₂]= 500ppm; [H₂O]=5 vol%; [O₂]=3 vol%.

As shown in Supplementary Figure 8, the NO_x conversions of Fe-Mn/Al-SBA-15 catalysts at 200 °C were initially 92.56%. It remarkably dropped to 67.58% after the treatment with 5% H₂O for 1 h. After that, the NO_x conversion further decreased to 64.79% in the presence of 500 ppm of SO₂ and 5% H₂O in 1 h. Because the denitrification performance of the Fe-Mn/Al-SBA-15 catalyst has decreased significantly in the presence of water and sulfur in 200 min, it is not necessary to perform the stability experiment for a longer time. Although the denitrification efficiency decreases intermediately after the treatment of H₂O and SO₂, it remains stable over the time. In industrial applications, the reduced airspeed may increase the denitrification efficiency. Moreover, the properties of the catalyst will be further optimized to improve its water and sulfur resistance and its stability.

3) As we all know, the hydrothermal stability of silica-aluminolite molecular sieve is not good. Has the author tested the hydrothermal stability of Al-SBA-15 molecular sieve?

Response: We thank the reviewer for the suggestions. We tested the hydrothermal stability of Al-SBA-15 molecular sieve by treating Al-SBA-15 MMS in a closed bottle at 100 °C for 300 h under static conditions. The samples after hydrothermal aging are marked as Al-SBA-15-ht. The high-temperature hydrothermal stability was tested by treating the sample at 600 °C for 6 h in a flow of 100% water vapor, which are termed as Al-SBA-15-hht. The structure and specific surface area of these samples were characterized by XRD and N₂ adsorption-desorption (see Supplementary Fig. 2 and Supplementary Fig. 3).

Supplementary Fig. 2 XRD patterns of Al-SBA-15 samples after different treatments

Supplementary Fig. 3 N₂ adsorption-desorption isotherms of Al-SBA-15 samples after different treatments

The results show that Al-SBA-15-ht and Al-SBA-15-hht both basically maintain the original morphology of Al-SBA-15 molecular sieve after hydrothermal treatment. The pore structure of Al-SBA-15-ht molecular sieve changed slightly. However, the

micropore structure of Al-SBA-15-hht molecular sieve disappeared, which suggests that the mesopore structure was partially destroyed after hydrothermal treatment at 600 °C. The higher skeletal silicon-aluminum ratio is responsible for the decrease of its high temperature hydrothermal stability. This is one of the reasons why mesoporous silicalite molecular sieves cannot be used in the petroleum catalytic cracking industry. Our results are consistent with previous studies.

Reviewer #2:

This manuscript reports on the development of novel low temperature catalysts for selective NH₃ catalytic reduction of NO_x pollutants. Conventional commercial catalysts do not work at flue temperatures below 300 °C, which heightens the importance of this study. Furthermore, the large process scale of reducing stationary NO_x emissions in power plants highlights the technological and economical significance of the work.

The study builds on previous work from the authors and focuses on the efficient doping of Al in SBA-15 mesoporous molecular sieves. Introduction of Al could induce the formation of Mn₂O₃ that enhances the selectivity to N₂. The authors rightly identify the importance of the acidity of the SBA-15 support and study the effects of pH as well as Si/Al ratio before they carry on to develop the optimal Fe-Mn/Al-SBA-15 catalyst. The authors have given a lot of thought in preparation of their catalyst development strategy, they used advanced techniques for catalyst sample characterisation, such as XRD, XPS, IR, TEM, NMR and TPD, as well as a continuous reactor system where they demonstrate the superior activity and selectivity of the developed catalyst. Although the study is mainly experimental, it is complemented by preliminary DFT theoretical calculations too.

Besides being well designed, the experimental work is carefully carried out and the results critically interpreted. The arguments of the authors are well presented, easy to follow and supported by literature references. The manuscript is well written with very important results and should be accepted as it is.

We would like to thank the reviewer for the positive comments.

Reviewer #3:

In this manuscript, the authors reported a novel catalytic system of Fe-Mn/Al-SBA-15 to reduce nitrogen monoxide with ammonia. The author discussed that Al in fly ash can guide the growth of Mn₂O₃ in the catalyst. This study sheds new light on the design and fabrication of novel catalytic system to solve the environmental crisis. However, to make

this manuscript stronger in terms of scientific importance, I believe this research article could be accepted for publication after revision. The detailed comments are as followings:

We would like to thank the reviewer for the valuable suggestions. Now we have carefully addressed the comments by correcting some sentences to support our results. Our point-to-point responses to reviewer's comments are listed below.

1) As we all know, the water resistance and sulfur resistance of the catalysts are also important in the reaction, please supplement the experiments.

Response: *Many thanks for the helpful suggestions. Reviewer 1 shared the same concern. We agree with both the reviewers that the water and sulfur resistance of the catalysts are very important. The extra experiments have been conducted to address this issue. The details can be found in the response of the comments of Review 1.*

2) Two different acid sites were described in the particle. Please explain briefly which acid site played a key role in this system. Some references about the acidic sites can increase catalytic activity should be cited and compared, such as *Angew. Chem.* (2019, 58, 6351); *Environ. Sci. Technol.* (2019, 53, 938–945) and others.

Response: *Many thanks for the suggestions. We have carefully read the articles recommended by the reviewer, which are very instructive to us. We cited the references recommended by reviewer marked as [23] and [24] in introduction in revised manuscript on page 2.*

*The paper published in *Angew. Chem.* reports that the incorporation of K^+ will significantly enhance the Lewis acid sites on the surface of $\alpha\text{-MnO}_2$ at low temperature, which can better adsorb and activate NH_3 molecules. This is crucial for its enhanced catalytic performance. Its NO_x conversion achieved 100 % at 150 °C, which is c.a. one time higher than that of pristine $\alpha\text{-MnO}_2$. The paper published in *Environ. Sci. Technol.* reports that the number of Brønsted acid sites is significantly increased with more active species produced by Fe_2O_3 promotion. In the presence of SO_2 , the Fe_2O_3 -promoted halloysite-supported $\text{CeO}_2\text{-WO}_3$ catalyst can effectively prevent the irreversible bonding of SO_2 with the active components, making the catalyst exhibit desirable sulfur resistance. The Fe_2O_3 -promoted halloysite-supported $\text{CeO}_2\text{-WO}_3$ catalyst exhibits superior catalytic activity, high N_2 selectivity over a wide temperature range from 270 to 420 °C, and excellent sulfur-poisoning resistance.*

Both papers suggest that the acidity of the active oxide component also has great influence on the denitrification activity. However, our research is focused on the acidity of the support. Based on our result, L acid and B acid coexist in molecular sieve support. The strength of L acid is greater than that of B acid. The spatial proximity of L acid and B acid enhances their synergistic effect and greatly enhanced their acidity. We agree that

L acid has a greater effect on low-temperature denitrification activity, while B acid has a greater effect on high-temperature denitrification activity. We believe that L acid sites can dominate catalytic reactions at 150-300□. The chemical nature of L acid sites is positively charged centres, which can attract the electrons to initiate the NH₃-SCR reaction.

3) In figure 4, whether the peaks of the known elements were optimally fitted, please explain briefly. And please add the title of the ordinate in the paper.

Response: *Yes, the XPS spectra of Fe 2p and Mn 2p in Fig. 4c, were optimally fitted according to the XRD result (Fig. 4d). The XRD pattern of our samples demonstrated that Mn₂O₃ and Mn₅O₈ with the lower valence state were formed after the introduction of aluminum to the molecular sieve. The XPS peak locations are consistent with the XRD results. Many thanks for the careful perusal. The title of the ordinate in XPS spectra has been added. We also repainted Fig. 4 in the revised manuscript accordingly.*

4) Grammar and expression should be reconsidered in some places.

Response: *Many thanks for the suggestion. We have invited Liwen Bianji from Edanz Group China (www.liwenbianji.cn/ac) to edit this manuscript.*

REVIEWERS' COMMENTS:

Reviewer #1 (Remarks to the Author):

I think this paper can be accepted as it now.

Reviewer #3 (Remarks to the Author):

After revision, the scientific significance of this paper had been improved. In my opinion, the revised manuscript could be accepted.